# PHASE-Net: Physics-Grounded Harmonic Attention System for Efficient Remote Photoplethysmography Measurement

## Abstract

Remote photoplethysmography (rPPG) measurement enables non-contact physiological monitoring but suffers from accuracy degradation under head motion and illumination changes. Existing deep learning methods are mostly heuristic and lack theoretical grounding, limiting robustness and interpretability. In this work, we propose a physics-informed rPPG paradigm derived from the Navier–Stokes equations of hemodynamics, showing that the pulse signal follows a second-order dynamical system whose discrete solution naturally leads to a causal convolution, justifying the use of a Temporal Convolutional Network (TCN). Based on this principle, we design the PHASE-Net, a lightweight model with three key components: 1) Zero-FLOPs Axial Swapper module to swap or transpose a few spatial channels to mix distant facial regions, boosting cross-region feature interaction without changing temporal order; 2) Adaptive Spatial Filter to learn a soft spatial mask per frame to highlight signal-rich areas and suppress noise for cleaner feature maps; and 3) Gated TCN, a causal dilated TCN with gating that models long-range temporal dynamics for accurate pulse recovery. Extensive experiments demonstrate that PHASE-Net achieves state-of-the-art performance and strong efficiency, offering a theoretically grounded and deployment-ready rPPG solution.

## 1 Introduction

Continuous monitoring of physiological signals, such as heart rate and heart rate variability, is fundamental to managing personal health and well-being. Traditional methods rely on contact-based sensors like ECG electrodes or pulse oximeters, which, despite their accuracy, are often inconvenient and uncomfortable for long-term, daily use. Remote photoplethysmography (rPPG) (**??**) has emerged as a revolutionary alternative, capable of reconstructing the pulse-wave signal from subtle, cardiac-induced variations in skin blood volume captured by a standard camera—all in a non-contact and imperceptible manner. This remarkable potential has positioned rPPG as a key enabling technology for a wide range of applications, including telemedicine, personal wellness tracking, driver monitoring, and affective computing (Chen et al., 2018; McDuff et al., 2014).

Despite its promise, the widespread adoption of rPPG in real-world scenarios faces significant hurdles. The core difficulty lies in the extremely faint nature of the physiological signal, which is easily overwhelmed by various noise sources (De Haan & Jeanne, 2013; Wang et al., 2017). For instance, involuntary head movements, facial expressions, and fluctuations in ambient illumination can introduce artifacts that are orders of magnitude stronger than the authentic pulse signal. To address these challenges, deep learning-based methods (Yu et al., 2019; Chen & McDuff, 2018; Yu et al., 2022) have become the dominant paradigm, demonstrating superior performance over traditional signal processing techniques by learning to regress the rPPG signal end-to-end from noisy video data.

However, we observe a fundamental limitation in the design philosophy of current deep learning models: they are, to a large extent, **heuristic**. Researchers typically frame rPPG as a generic spatio-temporal signal processing task, with network architectures often resulting from empirical trial-and-error. This **"black-box"** approach lacks a deep-seated understanding of the intrinsic physical laws governing the rPPG signal. This deficiency leads to two primary issues: 1) Models may overfit to dataset-specific noise patterns, resulting in poor generalization and a lack of robustness in unseen conditions, and 2) their poor interpretability makes it difficult to understand their decision-making process or guarantee their validity from a theoretical standpoint. This raises a critical question: Can we design an rPPG model whose architecture is a direct embodiment of the signal's physical principles, rather than merely a product of data fitting?

In this paper, to solve the above-mentioned issues, we introduce the **PHASE-Net (Physics-grounded Harmonic Attention System for Efficient rPPG measurement)**, a novel modeling framework rooted in the first principles of physics. Instead of treating the model as a black box, we begin with the Navier-Stokes equations for hemodynamics. Through a rigorous mathematical derivation, we reveal that the local pulse-wave dynamics can be physically described by a second-order damped harmonic oscillator model. Crucially, we further prove that the discrete-time solution to this physical model is formally equivalent to a causal convolution operator. This profound discovery provides an unequivocal theoretical justification for our use of a Temporal Convolutional Network (TCN) as the core dynamics modeling block, endowing our model with a powerful, physically-plausible inductive bias. The main contributions are summarized as follows:

- We propose a new rPPG modeling paradigm grounded in the first principles of physics and mathematics, for the first time establishing a theoretical bridge between the underlying physiological dynamics and a specific network architecture (causal convolution).
- We design a novel zero-FLOP module, **ZAS** (Zero-FLOPs Axial Swapper), which performs reversible spatial permutations on a small subset of channels to inject early cross-region interactions and strengthen long-range spatial dependencies without affecting the temporal axis.
- We introduce an **Adaptive Spatial Filtering (ASF)** module that not only generates a frame-wise spatial mask to highlight pulse-rich facial regions but also performs spatial aggregation and computes a first-order temporal derivative, concatenating it with the aggregated features to encode local pulse dynamics, thereby significantly enhancing model robustness under complex real-world conditions.
- Our final model, **PHASE-Net**, achieves state-of-the-art performance on multiple public datasets within an extremely lightweight architecture, demonstrating that theoretical rigor and practical efficiency can be achieved in unison.

## 2 RELATED WORK

### 2.1 TRADITIONAL SIGNAL PROCESSING BASED METHODS FOR rPPG MEASUREMENT

Early approaches typically extracted spatially averaged RGB traces from a facial region of interest (ROI) and applied Blind Source Separation (BSS) methods—such as ICA (Poh et al., 2010) or PCA (Lewandowska et al., 2011)—to separate the blood volume pulse (BVP) from noise. Building on skin–reflection priors, color–space designs such as CHROM (De Haan & Jeanne, 2013), POS (Wang et al., 2016), and 2SR (Wang et al., 2021) introduced specific projections or subspace rotations to enhance robustness against motion and illumination changes. These techniques established the foundation of rPPG research but rely on strong handcrafted assumptions and often break down under complex real-world motions or severe lighting variations.

### 2.2 DEEP LEARNING MODELS FOR rPPG MEASUREMENT

With the advent of deep learning, end-to-end networks have become dominant by directly learning spatio-temporal features from raw pixels and achieving large performance gains. 2D/3D CNNs such as DeepPhys (Chen & McDuff, 2018), PhysNet (Yu et al., 2019), and EfficientPhys (Liu et al., 2023) capture both spatial patterns and short-term dynamics but are computationally expensive and parameter-heavy. To better model long-range temporal dependencies, researchers have moved from CNN–RNN hybrids to Transformers (PhysFormer (Yu et al., 2022)) and selective state-space models (PhysMamba (Luo et al., 2024), RhythmMamba (Zou et al., 2025)) that enable linear-time sequence modeling with fine-grained temporal context. Most recently, PhysLLM (Xie et al., 2025) frames rPPG prediction as a language-like sequence modeling task, leveraging large language model backbones for stronger generalization. Despite their success, these architectures are largely borrowed from other domains and remain black-box, limiting interpretability and cross-domain robustness.

### 2.3 PHYSICS-INFORMED APPROACHES

Physics-Informed Neural Networks (PINNs) (Raissi et al., 2019) embed governing equations—typically partial differential equations—into the learning objective and have achieved remarkable success in fluid and solid mechanics by providing strong physical priors in data-scarce settings. In video-based physiological sensing, however, such principled integration of physics is still rare. Recent rPPG studies introduce periodic or contrastive physical losses (Choi & Lee, 2025; Sun & Li, 2024), but the network architectures themselves remain unconstrained by the underlying hemodynamics. Our proposed PHASE-Net differs fundamentally: starting from a hemodynamic formula-

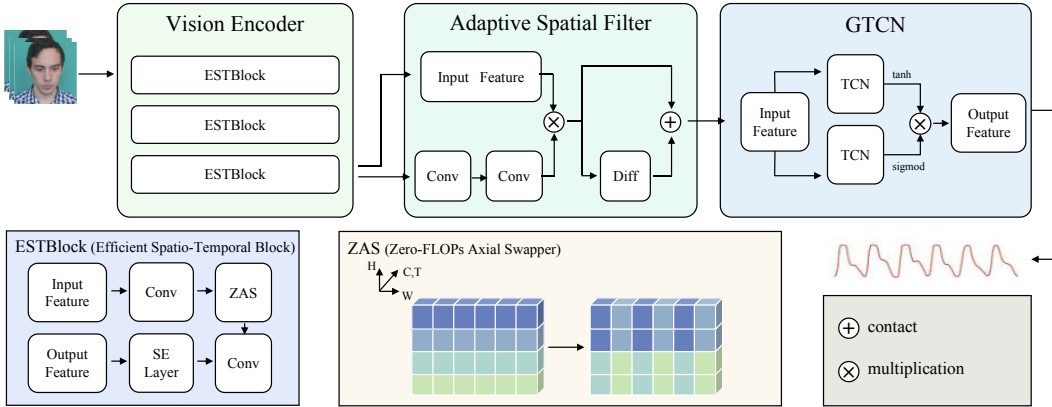

**Figure 1:** An overview of the PHASE-Net. The Vision Encoder comprises three Efficient Spatio-Temporal Blocks extracting spatio–temporal features from video inputs. These are fed into an Adaptive Spatial Filter module that computes filtered features via convolution layers and differential operations. The temporally refined features are then processed by a GTCN block, which uses dual-path Temporal Convolutional Networks with tanh and sigmoid gates for fusion. Also shown are the inner contents of ESTBlock (Efficient Spatio-Temporal Block) including ZAS (Zero-FLOPs Axial Swapper) that swaps spatial/temporal axes without adding FLOP.

tion, we derive a causal-convolution network whose computational structure is dictated by the physics itself, yielding a model that is both high-performing and intrinsically interpretable.

## 3 METHODOLOGY

### 3.1 PHYSICS-INFORMED TEMPORAL MODELING

Our proposed model, PHASE-Net, is founded on the principle that the neural network architecture for rPPG should serve as a parameterized approximation of the underlying physical laws of hemo-dynamics. This section details this principled approach. We first establish the link between visual observations and the latent physiological state. We then derive the governing physical law for this state and, finally, show its computational equivalence to our network architecture, which justifies our choice of a Temporal Convolutional Network (TCN).

#### 3.1.1 THE PHYSICAL OBSERVATION MODEL: FROM PIXELS TO LATENT STATE

Our derivation begins by establishing a link between the camera's visual signal and the physiological state of interest. This link is based on two principles: 1) The Beer-Lambert Law, which states that changes in captured pixel intensity $\Delta I(t)$ are proportional to changes in subcutaneous blood volume $\Delta V(t)$, and 2) Vessel Compliance, where $\Delta V(t)$ is proportional to the local blood pressure pulsation $\Delta p(t)$. We define this unobservable pressure pulsation as our target physical state, $z(t)$. This establishes a direct relationship:

$$z(t) \propto \Delta V(t) \propto \Delta I(t).$$

This physical relationship is the cornerstone of our methodology. It guarantees that the desired biological information, $z(t)$, is linearly encoded within the pixel value changes captured in the video stream $V$. The task of our visual encoder, $f_{enc}$, is therefore to disentangle and extract this information from the noisy observations to produce a feature estimate $z_{raw}$:

$$z_{raw} = f_{enc}(V) \approx z(t). \tag{1}$$

This estimate $z_{raw}$ is inevitably noisy. Our subsequent temporal model must leverage the physical laws governing $z(t)$ to purify this estimate.

#### 3.1.2 GOVERNING DYNAMICS: FROM FLUID DYNAMICS TO AN ODE

We now establish the dynamical equation that the 'clean' latent signal $z(t)$ must obey. We start from the Navier-Stokes equations, the most accurate physical description of blood flow:

$$\rho \left( \frac{\partial \mathbf{u}}{\partial t} + (\mathbf{u} \cdot \nabla)\mathbf{u} \right) = -\nabla p + \mu \nabla^2 \mathbf{u}, \tag{2}$$

$$\nabla \cdot \mathbf{u} = 0. \tag{3}$$

Given the intractability of this non-linear PDE system for our task, we introduce a series of physically-justified simplifications. First, we **linearize** the equations by considering the small pulsation component around the steady-state flow. Second, to model the collective effect in a skin patch, we perform **1D-averaging** along the pressure wave's principal axis. This yields a set of 1D linearized equations for momentum and continuity, where the viscous effects are modeled as a linear drag term $-ku'$ and vessel elasticity is incorporated via a compliance term $C$.

By combining these 1D equations and eliminating the velocity variable $u'$ (see Appendix C for detailed derivation), we arrive at a **Damped Wave Equation** that describes the propagation of the pressure pulse $p'$:

$$\frac{\partial^2 p'}{\partial t^2} + \alpha \frac{\partial p'}{\partial t} = c^2 \frac{\partial^2 p'}{\partial x^2}, \tag{4}$$

where $\alpha$ is a damping coefficient and $c$ is the wave speed. Crucially, the rPPG task involves a **single-point observation** at a fixed facial location ($x = x_0$). At this fixed point, the spatial derivative term $c^2 \frac{\partial^2 p'}{\partial x^2}$ represents the elastic restoring force from the surrounding tissue and fluid, which can be approximated as being proportional to the pressure deviation itself. This reduces the PDE to a classic second-order Ordinary Differential Equation (ODE), the **Forced Damped Harmonic Oscillator** model:

$$\frac{d^2 z(t)}{dt^2} + \alpha \frac{dz(t)}{dt} + \omega^2 z(t) = u(t), \tag{5}$$

Here, $z(t) := p'(x_0, t)$ is our latent signal, $\omega^2$ is the effective restoring force coefficient, and $u(t)$ represents external driving forces such as motion-induced noise. This ODE provides a powerful physical prior for the dynamics of any true rPPG signal.

### 3.1.3 Computational Equivalence: From an ODE to a TCN Architecture

The final step is to translate this physical law into a neural network architecture. We discretize the continuous ODE (Eq. 5) using a semi-implicit Euler method, which can be precisely represented as a Linear Time-Invariant State-Space Model :

$$\begin{aligned} \mathbf{x}_t &= \mathbf{A}\mathbf{x}_{t-1} + \mathbf{B}a_t, \\ z_t &= \mathbf{C}\mathbf{x}_t, \end{aligned} \tag{6}$$

where the state vector $\mathbf{x}_t = [z_t, v_t]^T$ contains the position and velocity of the oscillator, and the input $a_t$ is the discretized external force. The system matrices $(\mathbf{A}, \mathbf{B}, \mathbf{C})$ are determined entirely by the physical parameters $(\alpha, \omega)$ and the time step $\Delta t$.

Analyzing the solution to this state-space model leads to our core theoretical findings, which we formalize as two propositions.

**Proposition 1** (Equivalence to Causal Convolution). *The solution $z_t$ of the LTI system in Eq. 6 can be expressed as a causal convolution of all past inputs:*

$$z_t = \sum_{m=0}^{\infty} g[m] \cdot a_{t-m}, \quad \text{where} \quad g[m] = \mathbf{C}\mathbf{A}^m \mathbf{B}.$$

*Significance: This result rigorously transforms the physical model from a recursive form into a convolutional form, providing a theoretical basis for using a convolutional network to model the dynamics.*

**Proposition 2** (FIR Approximation). *The Infinite Impulse Response (IIR) convolution above can be approximated with arbitrary precision $\varepsilon$ by a Finite Impulse Response (FIR) filter of sufficient length, which is precisely the computation performed by a Temporal Convolutional Network (TCN).*

*Significance: This provides the final guarantee that a TCN is a principled architectural choice for implementing the physical dynamics of the rPPG signal with controllable error.*

These propositions form a complete logical chain from first principles to a specific network architecture. Therefore, the choice of a TCN in our PHASE-Net is not a heuristic one; it is the direct **architectural embodiment** of the physical laws governing the rPPG signal. Its role is to take the noisy feature estimate $z_{raw}$ and filter it such that the output conforms to this physically-mandated dynamical structure. Details can be seen in Appendix C. For theoretical guarantees of cross-domain generalization, please refer to Appendix D.

## 3.2 Zero-FLOPs Axial Swapper

The **Zero-FLOPs Axial Swapper (ZAS)** is a lightweight, parameter-free operator that introduces early cross-region interactions with *zero* computational cost. It performs a reversible block-wise spatial transpose on a small subset of channels while strictly preserving the temporal dimension, providing richer spatial dependencies for subsequent physics-informed temporal modeling.

**Mathematical Definition.** Let the input feature map be

$$X \in \mathbb{R}^{B \times C \times T \times H \times W}, \tag{7}$$

where $B$ is the batch size, $C$ the channel dimension, $T$ the temporal length, and $H, W$ the spatial dimensions. ZAS acts only on the last $k = \lfloor pC \rfloor$ channels ($0 < p < 1$), leaving the remaining $C - k$ channels unchanged:

$$X = \begin{bmatrix} X_{\text{id}}, \ X_{\text{swap}} \end{bmatrix}, \qquad X_{\text{id}} \in \mathbb{R}^{B \times (C-k) \times T \times H \times W}. \tag{8}$$

Given a block size $b$, each spatial slice of $X_{\text{swap}}$ is partitioned into non-overlapping $b \times b$ blocks

$$\mathcal{P} : \ \mathbb{R}^{H \times W} \rightarrow \mathbb{R}^{\frac{H}{b} \times \frac{W}{b} \times b \times b}, \tag{9}$$

and a two-dimensional transpose is applied inside every block

$$\mathcal{T}(Z)_{u,v} = Z_{v,u}, \quad Z \in \mathbb{R}^{b \times b}. \tag{10}$$

The overall ZAS transformation is

$$\text{ZAS}(X_{\text{swap}}) = \mathcal{P}^{-1}\big(\mathcal{T}(\mathcal{P}(X_{\text{swap}}))\big), \qquad \tilde{X} = \begin{bmatrix} X_{\text{id}}, \ \text{ZAS}(X_{\text{swap}}) \end{bmatrix}. \tag{11}$$

**Theoretical Properties**

**Proposition 3** (Self-inversion)**.**

$$\text{ZAS}(\text{ZAS}(X_{\text{swap}})) = X_{\text{swap}}.$$

*This property guarantees that ZAS is a* complete and reversible mapping*, which ensures feature consistency and stable gradient propagation even when ZAS is repeatedly applied in deep networks.*

**Proposition 4** (Energy preservation and 1-Lipschitz)**.** *Because both $\mathcal{P}$ and $\mathcal{T}$ are pure permutations,*

$$\|\text{ZAS}(X_{\text{swap}})\|_2 = \|X_{\text{swap}}\|_2, \qquad \text{Lip}(\text{ZAS}) = 1.$$

*The output norm exactly matches the input norm, preventing signal amplification or attenuation and improving training stability.*

**Complexity.** ZAS introduces no learnable parameters and no multiply–accumulate operations, resulting in theoretical FLOPs of **0** and parameter count of **0**. Its runtime cost is dominated by indexing, with time complexity

$$O(B \cdot k \cdot T \cdot H \cdot W).$$

The detailed description of the ZAS module is provided in the Appendix E.

## 3.3 Adaptive Spatial Filter

The feature representations learned from video for rPPG are inherently subject to the challenge of **spatial heterogeneity**. The target physiological signal exhibits a high signal-to-noise ratio (SNR) only in specific facial regions (e.g., the forehead and cheeks), while other areas are dominated by irrelevant **nuisance variations**, such as non-rigid deformations from facial expressions and specular reflections under changing illumination. In this context, a naive aggregation operator like Global Average Pooling (GAP), which imposes a **uniform prior** over all spatial locations, is suboptimal and inevitably produces corrupted temporal features where signal-bearing patterns are contaminated by these nuisance variations.

To address this challenge, we introduce a learnable, dynamic spatial filtering mechanism called the **Adaptive Spatial Filter (ASF)**, which adaptively aggregates information from the high-dimensional feature map and further enriches the representation by explicitly encoding temporal dynamics. Given spatio-temporal features $Z \in \mathbb{R}^{B \times C' \times T \times H \times W}$ from the visual encoder, ASF first estimates an unnormalized spatial logit map $M'_t \in \mathbb{R}^{B \times 1 \times H \times W}$ for each frame $t$ via a lightweight convolutional network $f_{conv}$:

$$M'_t = f_{conv}(Z_{:,:,t}). \tag{12}$$

The logits are converted into a normalized attention mask $M_t$ through a spatial Softmax:

$$\text{vec}(M_t) = \text{softmax}\big(\text{vec}(M_t')\big), \tag{13}$$

where $\text{vec}(\cdot)$ flattens the spatial dimensions $(H, W)$. This mask assigns higher weights to signal-rich regions and lower weights to noisy ones. The weighted feature for each frame is then obtained by

$$\hat{Z}_t = Z_{:,:,t} \odot M_t, \tag{14}$$

where $\odot$ denotes element-wise multiplication with broadcasting. Aggregating over the spatial dimensions yields a robust 1D feature vector

$$\mathbf{z}_t = \sum_{h,w} \hat{Z}_{t,:,h,w}. \tag{15}$$

To explicitly capture the local temporal dynamics of the rPPG signal, ASF further computes the **first-order temporal derivative** of the aggregated sequence:

$$\mathbf{v}_t = \mathbf{z}_t - \mathbf{z}_{t-1}, \qquad t = 2, \ldots, T, \tag{16}$$

where $\mathbf{v}_t$ represents the instantaneous "velocity'' of the latent pulse representation. The final ASF output is formed by channel-wise concatenation of the static aggregated feature and its temporal derivative,

$$\mathbf{z}_t' = [\, \mathbf{z}_t, \, \mathbf{v}_t \,], \tag{17}$$

which preserves both the spatially purified intensity and the short-term temporal variation of the underlying blood volume pulse.

From a **representation learning** perspective, ASF acts as a **disentangling** mechanism. It collapses the noisy spatial dimensions while simultaneously encoding instantaneous temporal changes, yielding a low-dimensional but high-fidelity sequence that serves as an ideal input for the downstream physics-informed temporal model. By providing both clean spatial aggregation and explicit motion-aware dynamics, ASF enables the physical model to focus on fitting the intrinsic hemodynamic patterns rather than combating confounding visual noise, thereby improving accuracy and generalization.

### 3.4 TRAINING OBJECTIVE

The primary training objective $\mathcal{L}_{\text{pred}}$ for the proposed PHASE-Net is to maximize the morphological similarity between the predicted rPPG waveform $\hat{\mathbf{y}} \in \mathbb{R}^T$ and the ground truth signal $\mathbf{y} \in \mathbb{R}^T$. We employ a Negative Pearson Correlation loss, which directly optimizes this objective and is a strong standard for physiological signal regression:

$$\mathcal{L}_{\text{pred}} = -\frac{\sum_{t=1}^{T} (\hat{y}_t - \bar{\hat{\mathbf{y}}})(y_t - \bar{\mathbf{y}})}{\sqrt{\sum_{t=1}^{T} (\hat{y}_t - \bar{\hat{\mathbf{y}}})^2 \sum_{t=1}^{T} (y_t - \bar{\mathbf{y}})^2}}, \tag{18}$$

where $\bar{\hat{\mathbf{y}}}$ and $\bar{\mathbf{y}}$ denote the mean values of the predicted and ground truth signals, respectively.

## 4 EXPERIMENTS

We evaluate on UBFC-rPPG (Bobbia et al., 2017), PURE (Stricker et al., 2014), BUAA-MIHR/BUAA (Xi et al., 2020), and MMPD (Tang et al., 2023) under standard intra-dataset and cross-dataset protocols. Dataset descriptions and implementation details are in Appendix A and B.

### 4.1 INTRA-DATASET EVALUATION

We first evaluate PHASE-Net on the standard intra-dataset benchmark, where the model is trained and tested on splits from the same dataset to measure predictive power under consistent conditions. The detailed results are presented in Table 1. Across all four benchmarks, PHASE-Net delivers the lowest or near-lowest errors and the highest correlations. On UBFC-rPPG, our method achieves an MAE of 0.15 bpm and RMSE of 0.53 bpm with $R = 0.99$, surpassing the previous best MAE of 0.16 bpm by LST-rPPG and demonstrating excellent waveform fidelity. On PURE, PHASE-Net attains a remarkable 0.14 bpm MAE and 0.35 bpm RMSE while maintaining $R = 0.99$, cutting the MAE by roughly half compared with strong recent baselines such as RhythmFormer (0.27 bpm) or PhysDiff (0.29 bpm). Even on the more challenging BUAA dataset, which features significant illumination

changes and device diversity, our model achieves 5.89 bpm MAE and 7.89 bpm RMSE with a positive correlation of 0.48; competing deep models such as PhysFormer suffer negative correlations and considerably higher errors. On MMPD, which introduces diverse sensors and colored lighting, PHASE-Net reaches 4.78 bpm MAE and 8.22 bpm RMSE with $R = 0.71$, again outperforming all baselines and preserving temporal structure despite domain complexity. These results highlight that PHASE-Net delivers low errors across both controlled (UBFC, PURE) and complex (BUAA, MMPD) settings, with high correlations ensuring faithful waveform recovery for downstream analysis. Its physics-driven causal convolution, adaptive spatial filter, and parameter-free ZAS module together enable these gains with only 0.29 M parameters, achieving strong accuracy, robustness, and efficiency.

Additional qualitative examples of predicted versus ground-truth rPPG signals are provided in Appendix G, where the waveform and PSD plots further illustrate the fidelity of PHASE-Net's predictions .

**Table 1:** Intra-dataset evaluation on UBFC-rPPG, PURE, BUAA and MMPD datasets. Best results are in **bold**.

| Method | UBFC-rPPG | | | PURE | | | BUAA | | | MMPD | | |
|---|---|---|---|---|---|---|---|---|---|---|---|---|
| | MAE↓ | RMSE↓ | R↑ | MAE↓ | RMSE↓ | R↑ | MAE↓ | RMSE↓ | R↑ | MAE↓ | RMSE↓ | R↑ |
| Green (Verkruysse et al., 2008) | 19.73 | 31.00 | 0.37 | 10.09 | 23.85 | 0.34 | 6.89 | 10.39 | 0.60 | 21.68 | 27.69 | -0.01 |
| ICA (Poh et al., 2010) | 16.00 | 25.65 | 0.44 | 4.77 | 16.07 | 0.72 | - | - | - | 18.60 | 24.30 | 0.01 |
| CHROM (De Haan & Jeanne, 2013) | 4.06 | 8.83 | 0.89 | 5.77 | 14.93 | 0.81 | - | - | - | 13.66 | 18.76 | 0.08 |
| POS (Wang et al., 2016) | 4.08 | 7.72 | 0.92 | 3.67 | 11.82 | 0.88 | - | - | - | 12.36 | 17.71 | 0.18 |
| PhysNet (Yu et al., 2019) | 2.95 | 3.67 | 0.97 | 2.10 | 2.60 | 0.99 | 10.89 | 11.70 | -0.04 | 4.80 | 11.80 | 0.60 |
| Meta-rPPG (Lee et al., 2020) | 5.97 | 7.42 | 0.57 | 2.52 | 4.63 | 0.98 | - | - | - | - | - | - |
| PhysFormer (Yu et al., 2022) | 0.92 | 2.46 | 0.99 | 1.10 | 1.75 | 0.99 | 8.45 | 10.17 | -0.06 | 11.99 | 18.41 | 0.18 |
| EfficientPhys (Liu et al., 2023) | 1.41 | 1.81 | 0.99 | 4.75 | 9.39 | 0.99 | 16.09 | 16.80 | 0.14 | 13.47 | 21.32 | 0.21 |
| Contrast-Phys+ (Sun & Li, 2024) | 0.21 | 0.80 | 0.99 | 0.48 | 0.98 | 0.99 | - | - | - | - | - | - |
| DiffPhys (Chen et al., 2024) | 1.05 | 1.63 | 0.99 | 1.46 | 5.88 | 0.90 | - | - | - | - | - | - |
| RhythmFormer (Zou et al., 2025) | 0.50 | 0.78 | 0.99 | 0.27 | 0.47 | 0.99 | 9.19 | 11.93 | -0.10 | **4.69** | 11.31 | 0.60 |
| STFPNet (Li et al., 2025b) | 0.41 | 0.95 | 0.99 | 0.47 | 0.67 | 0.99 | - | - | - | - | - | - |
| Style-rPPG (Liu et al., 2025) | 0.17 | 0.41 | 0.99 | 0.39 | 0.62 | 0.99 | - | - | - | - | - | - |
| LST-rPPG (Li et al., 2025a) | 0.16 | 0.57 | 0.99 | 0.32 | 0.62 | 0.99 | - | - | - | - | - | - |
| PhysDiff (Qian et al., 2025) | 0.33 | 0.57 | 0.99 | 0.29 | 0.54 | 0.99 | - | - | - | 7.17 | 9.63 | 0.78 |
| **PHASE-Net (Ours)** | **0.15** | **0.53** | **0.99** | **0.14** | **0.35** | **0.99** | **5.89** | **7.89** | **0.48** | 4.78 | **8.22** | **0.71** |

## 4.2 Generalization Ability Evaluation

**Multi-Domain Generalization.** We evaluate PHASE-Net using the leave-one-out protocol, training on three datasets and testing on the remaining one to simulate deployment in unseen environments and rigorously assess domain invariance.

As shown in Table 2, PHASE-Net achieves the best overall performance on all four transfer directions, often by a large margin. When transferring to PURE, it records 2.86 bpm MAE and 9.66 bpm RMSE with $R = 0.91$, outperforming the next best deep model RhythmFormer (21.11/25.76) by over an order of magnitude. For BUAA with severe illumination variation, it attains 2.56 bpm MAE and 3.25 bpm RMSE with $R = 0.96$, whereas PhysFormer shows errors above 22 bpm and near-zero correlation. Even in the more moderate UBFC and MMPD transfers, PHASE-Net remains superior: 10.04/15.56 bpm MAE/RMSE ($R = 0.65$) on UBFC and 10.33/16.20 bpm ($R = 0.40$) on MMPD, outperforming both classical signal-processing baselines and recent deep networks.

These results confirm that PHASE-Net learns physics-aligned representations rather than dataset-specific appearance cues, providing stable predictive power and strong cross-domain robustness even when the target domain differs greatly from the training distributions. The combination of a causal convolution derived from hemodynamic principles, an adaptive spatial filter that focuses on signal-rich regions, and the parameter-free ZAS module collectively reinforces temporal consistency and prevents overfitting to superficial domain artifacts.

**Limited-Source Domain Generalization.** We further evaluate a limited-source setting where the model is trained on only two datasets and tested on a third unseen target domain, simulating deployment with scarce and heterogeneous training data. Table 3 shows that PHASE-Net consistently

**Table 2:** Multi-domain generalization evaluation (Leave-One-Out Protocol). U=UBFC-rPPG, P=PURE, B=BUAA-MIHR, M=MMPD. Best results are marked in **bold**.

| Method | Other→U | | | Other→P | | | Other→B | | | Other→M | | |
|---|---|---|---|---|---|---|---|---|---|---|---|---|
| | MAE↓ | RMSE↓ | R↑ | MAE↓ | RMSE↓ | R↑ | MAE↓ | RMSE↓ | R↑ | MAE↓ | RMSE↓ | R↑ |
| Green (Verkruysse et al., 2008) | 19.73 | 31.00 | 0.37 | 10.09 | 23.85 | 0.34 | 6.89 | 10.39 | 0.60 | 21.68 | 27.69 | -0.01 |
| CHROM (De Haan & Jeanne, 2013) | 7.23 | 8.92 | 0.51 | 9.79 | 12.76 | 0.37 | 6.09 | 8.29 | 0.51 | 13.66 | 18.76 | 0.08 |
| POS (Wang et al., 2016) | 7.35 | 8.04 | 0.49 | 9.82 | 13.44 | 0.34 | 5.04 | 7.12 | 0.63 | 12.36 | 17.71 | 0.18 |
| EfficientPhys (Liu et al., 2023) | 12.87 | 18.80 | 0.19 | 7.15 | 15.04 | 0.23 | 32.30 | 34.00 | -0.03 | 12.87 | 18.80 | 0.19 |
| PhysFormer (Yu et al., 2022) | 10.29 | 18.13 | 0.60 | 19.75 | 24.30 | 0.24 | 22.09 | 26.21 | 0.03 | 13.90 | 19.30 | 0.06 |
| PhysNet (Yu et al., 2019) | 13.83 | 23.66 | 0.35 | 33.23 | 35.25 | -0.15 | 12.75 | 16.37 | 0.08 | 13.37 | 16.64 | 0.29 |
| RhythmFormer (Zou et al., 2025) | 14.71 | 22.49 | 0.43 | 21.11 | 25.76 | 0.04 | 6.04 | 10.84 | 0.42 | 16.14 | 20.50 | -0.11 |
| **PHASE-Net (Ours)** | **10.04** | **15.56** | **0.65** | **2.86** | **9.66** | **0.91** | **2.56** | **3.25** | **0.96** | **10.33** | **16.20** | **0.40** |

achieves the best or near-best results across all source–target pairs. When trained on PURE+UBFC and tested on the challenging MMPD, our model reaches an MAE of **9.76** bpm and RMSE of 16.07 bpm ($R = 0.39$), outperforming RhythmFormer and other deep baselines. Training on PURE+BUAA yields similar gains, with MAE/RMSE of 11.38/15.96 bpm, while generalization to the illumination-sensitive BUAA dataset is especially strong: using PURE+UBFC as sources, PHASE-Net lowers the MAE to 2.91 bpm and RMSE to 4.23 bpm with a correlation of 0.92, well ahead of all competitors. These results confirm that by leveraging physics-grounded modeling, PHASE-Net captures domain-invariant physiological dynamics rather than overfitting to superficial dataset biases.

**Table 3:** Results of limited-source domain generalization on MMPD (left) and BUAA-MIHR (right).

| Train | Model | MAE | RMSE | R | Train | Model | MAE | RMSE | R |
|---|---|---|---|---|---|---|---|---|---|
| | Green (Verkruysse et al., 2008) | 21.68 | 27.69 | -0.01 | | Green (Verkruysse et al., 2008) | 6.89 | 10.39 | 0.60 |
| | PhysNet (Yu et al., 2019) | 13.2 | 16.7 | 0.23 | | PhysNet (Yu et al., 2019) | 20.97 | 24.75 | 0.01 |
| PURE+BUAA | PhysFormer (Yu et al., 2022) | 13.9 | 18.6 | 0.21 | PURE+MMPD | PhysFormer (Yu et al., 2022) | 14.86 | 18.26 | 0.03 |
| | EfficientPhys (Liu et al., 2023) | 11.9 | 18.5 | 0.21 | | EfficientPhys (Liu et al., 2023) | 4.15 | 7.14 | 0.77 |
| | RhythmFormer (Zou et al., 2025) | 13.98 | 19.46 | 0.12 | | RhythmFormer (Zou et al., 2025) | 4.32 | 6.70 | 0.82 |
| | **PHASE-Net (Ours)** | **11.38** | **15.96** | **0.30** | | **PHASE-Net (Ours)** | **4.03** | **6.21** | **0.85** |
| | Green (Verkruysse et al., 2008) | 21.68 | 27.69 | -0.01 | | Green (Verkruysse et al., 2008) | 6.89 | 10.39 | 0.60 |
| | PhysNet (Yu et al., 2019) | 11.0 | 17.3 | 0.28 | | PhysNet (Yu et al., 2019) | 11.40 | 16.72 | 0.14 |
| PURE+UBFC | PhysFormer (Yu et al., 2022) | 11.4 | 17.5 | 0.23 | MMPD+UBFC | PhysFormer (Yu et al., 2022) | 10.87 | 16.20 | 0.08 |
| | EfficientPhys (Liu et al., 2023) | 11.8 | 18.9 | 0.22 | | EfficientPhys (Liu et al., 2023) | 3.00 | 5.18 | 0.89 |
| | RhythmFormer (Zou et al., 2025) | 10.50 | 16.72 | 0.28 | | RhythmFormer (Zou et al., 2025) | 6.20 | 11.23 | 0.49 |
| | **PHASE-Net (Ours)** | **9.76** | **16.07** | **0.39** | | **PHASE-Net (Ours)** | **3.51** | **5.18** | **0.89** |
| | Green (Verkruysse et al., 2008) | 21.68 | 27.69 | -0.01 | | Green (Verkruysse et al., 2008) | 6.89 | 10.39 | 0.60 |
| | PhysNet (Yu et al., 2019) | 13.5 | 17.0 | 0.09 | | PhysNet (Yu et al., 2019) | 15.34 | 21.48 | -0.29 |
| BUAA+UBFC | PhysFormer (Yu et al., 2022) | 13.2 | 16.5 | 0.12 | PURE+UBFC | PhysFormer (Yu et al., 2022) | 18.23 | 22.17 | 0.07 |
| | EfficientPhys (Liu et al., 2023) | 15.5 | 20.8 | 0.03 | | EfficientPhys (Liu et al., 2023) | 4.60 | 8.06 | 0.72 |
| | RhythmFormer (Zou et al., 2025) | 12.57 | 17.45 | 0.15 | | RhythmFormer (Zou et al., 2025) | 3.90 | 6.51 | 0.82 |
| | **PHASE-Net (Ours)** | **11.84** | **17.47** | **0.15** | | **PHASE-Net (Ours)** | **2.91** | **4.23** | **0.92** |

**Efficiency Analysis.** We compare both parameter counts and multiply–accumulate operations (MACs) in Table 4. Under a 128×128 spatial resolution and $T$=128 frames per clip, PHASE-Net requires only 0.29M parameters and 28.3G MACs, notably lower than most prior arts while maintaining state-of-the-art accuracy.This lightweight design enables faster inference and easier deployment on edge devices without sacrificing cross-domain robustness.

### 4.3 Ablation Study

**Study of Different Modules.** Under the same training and evaluation settings as the main results, we remove one module at a time from PHASE-Net and report RMSE results on UBFC-rPPG and PURE datasets (see Fig. 2a). The full model reaches 0.90 bpm on UBFC-rPPG and 0.14 bpm on PURE. On UBFC-rPPG, the largest degradation

**Table 4:** Efficiency analysis.

| Method | Param. (M) | MACs (G) |
|---|---|---|
| TS-CAN | 7.50 | 96.0 |
| PhysNet | 0.77 | 56.1 |
| DeepPhys | 7.50 | 96.0 |
| EfficientPhys | 7.40 | 45.6 |
| PhysFormer | 7.38 | 40.5 |
| RhythmFormer | 4.21 | 28.8 |
| Contrast-Phys+ | 0.85 | 145.7 |
| PhysMamba | 0.56 | 47.3 |
| **MDNet (Ours)** | **0.29** | **28.3** |

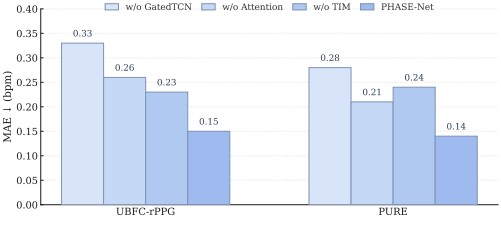
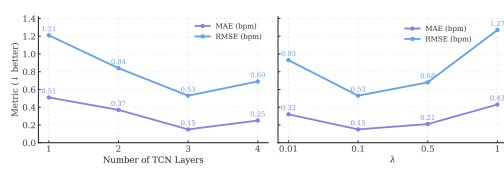

(a) Module ablation study.

(b) Ablation of TCN layers and $\lambda$ on UBFC-rPPG.

**Figure 2:** Comparison of different ablation studies.

appears when removing GTCN: 0.90→1.26 bpm; removing Attention is also detrimental, while removing ZAS yields a smaller increase about 0.14 bpm. On PURE, Attention is the most critical: 0.14→0.36 bpm; ZAS and GTCN also help but with smaller margins.

Ablation studies reveal that all component removals degrade performance, highlighting their complementary roles. Attention is most critical in scenarios with strong local artifacts. The GTCN module contributes significantly by capturing longer-range rhythmic stability, while the ZAS module provides low-cost early temporal alignment, yielding consistent gains. Our full model, by combining these modules, achieves the lowest error across all evaluation scenarios.

**Depth of the TCN backbone.** We vary the number of TCN layers from 1 to 4 and evaluate on UBFC-rPPG (Fig. 2b left). Performance consistently improves when increasing the depth from 1 to 3 layers: MAE drops from 0.51 to 0.15 bpm (70.6% relative reduction) and RMSE from 1.21 to 0.53 bpm (56.2%). Adding a fourth layer slightly degrades the accuracy (MAE/RMSE = 0.25/0.69). We hypothesize that three layers provide a sufficient temporal receptive field for pulse dynamics, while deeper stacks start to over-smooth and complicate optimization. Therefore, we set the default depth to 3.

**Effect of the loss weight $\lambda$.** We sweep $\lambda \in \{0.01, 0.1, 0.5, 1\}$ to balance training objectives (Fig. 2b right). A clear U-shaped trend is observed: $\lambda = 0.1$ achieves the best trade-off with MAE/RMSE = 0.15/0.53 bpm. Compared to $\lambda = 0.01$, this setting reduces MAE by 53.1% and RMSE by 43.0%. Increasing $\lambda$ beyond 0.1 over-regularizes the model (e.g., $\lambda = 1$: 0.43/1.27), while a too small weight under-utilizes the auxiliary objective (0.32/0.93 at $\lambda = 0.01$). Unless stated otherwise, we use $\lambda = 0.1$ in all experiments.

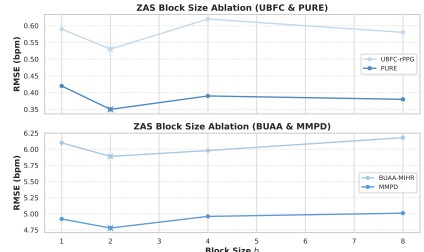

**Figure 3:** Ablation over ZAS block sizes $b$.

**ZAS Ablation.** We further investigate the influence of ZAS hyper-parameters by varying both the spatial block size $b$ and the number of swapped channel groups $p_c$. As shown in Fig. 3 and Fig. 4, performance consistently peaks at $b = 2$ and $p_c = 2$. A fine-grained 2×2 spatial permutation provides sufficient cross-region mixing while preserving local structures, and a moderate channel-group swap delivers the strongest cross-domain robustness. These results confirm that ZAS enhances generalization primarily through balanced spatial interaction rather than aggressive reordering.

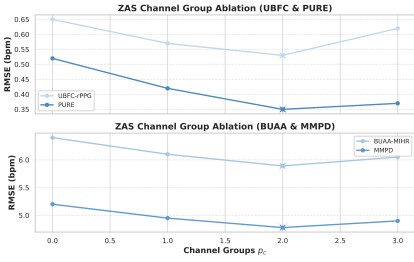

**Figure 4:** Ablation over ZAS channel groups $p_c$.

## 5 CONCLUSION

In this paper, we introduced PHASE-Net, a physics-grounded rPPG model that embodies a damped harmonic oscillator through a causal (finite) convolution. The design couples an adaptive spatial filter and a Zero-FLOPs Axial Swapper (ZAS) with a compact GTCN. Experiments demonstrate a strong balance of accuracy, cross-domain robustness, and efficiency. We hope this work encourages moving from heuristic stacking toward principled, task-specific inductive biases for modeling physiological signals from video. Building on this foundation, future work can explore extending the physics-based formulation to multi-task physiological sensing, such as respiration or blood pressure. Moreover, the modular nature of PHASE-Net makes it readily adaptable to other video-based biomedical applications where interpretability and domain generalization are critical.

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

## A   INTRODUCTION TO THE DATASETS

**UBFC-rPPG Bobbia et al. (2017)** contains 42 RGB facial videos from 42 distinct subjects. Each video is captured at 640×480 pixel resolution and 30 frames per second (fps). Recordings take place under varied lighting conditions, including natural sunlight and indoor artificial illumination. Ground-truth physiological signals are recorded via a CMS50E pulse oximeter at 60 Hz, ensuring precise temporal alignment for evaluation.

**PURE Stricker et al. (2014)** comprises 60 high-quality RGB videos collected from 10 subjects performing six different head movement scenarios (static, talking, translation movements, etc.). Videos are recorded at 30 fps under consistent indoor lighting and controlled background settings, minimizing external interference. Synchronized physiological measurements are obtained using a CMS50E

oximeter sampling at 60 Hz. PURE is particularly valuable for evaluating rPPG performance during facial movements.

**BUAA Xi et al. (2020)** is designed to assess algorithmic robustness across varying illumination intensities. The dataset features video sequences recorded under a range of controlled lighting conditions, from low-light (below 10 lux) to normal brightness. In our experiments, we only utilize videos captured under illumination levels ≥10 lux, as extremely dim lighting introduces significant image degradation requiring specialized enhancement techniques beyond this study's scope.

**MMPD Tang et al. (2023)** comprises 660 videos, each lasting one minute, collected from 33 subjects with diverse skin tones and gender distributions. Each video is recorded at 30 fps with a resolution of 320×240 pixels, under four distinct lighting conditions (bright, warm, dim, and colored lighting). Subjects perform various daily activities, introducing intra-subject variability and further increasing dataset complexity.

## B  Implementation Details

Our PHASE-Net is implemented using PyTorch. The input to the network is a sequence of 128 frames, resized to $128 \times 128$. We trained the model for 15 epochs using the Adam optimizer with a learning rate of $10^{-4}$ and a batch size of 4. The loss function hyperparameter was set to $\lambda = 0.1$. All experiments were conducted on a single NVIDIA H100 GPU.

## C  Detailed Derivation of the Physics-Informed Temporal Model

This appendix provides the detailed mathematical derivations for the physics-informed temporal model, as summarized in Section 3.1.

### C.1  Derivation of the Damped Wave Equation (PDE)

Our goal is to derive a single equation for the pressure pulsation $p'$ from the 1D linearized equations for momentum and continuity:

$$\rho \frac{\partial u'}{\partial t} + k u' = -\frac{\partial p'}{\partial x} \tag{19}$$

$$\frac{\partial Q'}{\partial x} = -C \frac{\partial p'}{\partial t} \tag{20}$$

where $Q' = Au'$ is the flow rate, and $A$ is the cross-sectional area of the vessel. The derivation proceeds in the following steps:

1. We take the partial derivative of the momentum equation (Eq. 19) with respect to the spatial variable $x$:

$$\frac{\partial}{\partial x}\left(\rho \frac{\partial u'}{\partial t} + k u'\right) = \frac{\partial}{\partial x}\left(-\frac{\partial p'}{\partial x}\right)$$

Assuming fluid properties $\rho, k$ are locally uniform and swapping the order of differentiation, we get:

$$\rho \frac{\partial}{\partial t}\left(\frac{\partial u'}{\partial x}\right) + k \left(\frac{\partial u'}{\partial x}\right) = -\frac{\partial^2 p'}{\partial x^2} \tag{21}$$

2. We relate the velocity gradient $\frac{\partial u'}{\partial x}$ to the flow rate gradient $\frac{\partial Q'}{\partial x}$. Since $Q' = Au'$, under the small pulsation assumption, the area $A$ can be approximated by its mean value $\bar{A}$, so $Q' \approx \bar{A}u'$. Taking the spatial derivative yields:

$$\frac{\partial u'}{\partial x} \approx \frac{1}{\bar{A}} \frac{\partial Q'}{\partial x} \tag{22}$$

3. We substitute Eq. 22 into Eq. 21 to replace the velocity gradient with the flow rate gradient:

$$\rho \frac{\partial}{\partial t}\left(\frac{1}{\bar{A}} \frac{\partial Q'}{\partial x}\right) + \frac{k}{\bar{A}}\left(\frac{\partial Q'}{\partial x}\right) = -\frac{\partial^2 p'}{\partial x^2}$$

4. Finally, we use the continuity equation (Eq. 20) to replace the flow rate gradient term $\frac{\partial Q'}{\partial x}$ with the pressure term $-C \frac{\partial p'}{\partial t}$:

$$\frac{\rho}{\bar{A}} \frac{\partial}{\partial t}\left(-C \frac{\partial p'}{\partial t}\right) + \frac{k}{\bar{A}}\left(-C \frac{\partial p'}{\partial t}\right) = -\frac{\partial^2 p'}{\partial x^2}$$

Rearranging the terms, we obtain:

$$\frac{\rho C}{\bar{A}} \frac{\partial^2 p'}{\partial t^2} + \frac{kC}{\bar{A}} \frac{\partial p'}{\partial t} = \frac{\partial^2 p'}{\partial x^2}$$

5. By defining new physical constants for wave speed squared ($c^2 := \frac{\bar{A}}{\rho C}$) and a damping-related coefficient, we arrive at the final Damped Wave Equation presented in the main text:

$$\frac{\partial^2 p'}{\partial t^2} + \alpha \frac{\partial p'}{\partial t} = c^2 \frac{\partial^2 p'}{\partial x^2} \tag{23}$$

## C.2 Discretization and State-Space Formulation

We start with the second-order ODE for the damped harmonic oscillator:

$$\frac{d^2 z(t)}{dt^2} + \alpha \frac{dz(t)}{dt} + \omega^2 z(t) = u(t) \tag{24}$$

First, we convert this into a system of two first-order ODEs by defining the state vector $\mathbf{x}(t) = [z(t), v(t)]^T$, where $v(t) = \frac{dz(t)}{dt}$ is the velocity.

$$\frac{dz(t)}{dt} = v(t)$$

$$\frac{dv(t)}{dt} = -\alpha v(t) - \omega^2 z(t) + u(t)$$

We discretize this system using a semi-implicit Euler method with a time step $\Delta t$. Let $z_t \approx z(t\Delta t)$ and $a_t \approx u(t\Delta t)$. The update rules are:

$$v_t = v_{t-1} + \Delta t \cdot (-\alpha v_t - \omega^2 z_{t-1} + a_t) \tag{25}$$

$$z_t = z_{t-1} + \Delta t \cdot v_t \tag{26}$$

We first solve for $v_t$ from Eq. 25:

$$(1 + \alpha \Delta t)v_t = v_{t-1} - \omega^2 \Delta t z_{t-1} + \Delta t a_t$$

$$v_t = \frac{1}{1 + \alpha \Delta t} v_{t-1} - \frac{\omega^2 \Delta t}{1 + \alpha \Delta t} z_{t-1} + \frac{\Delta t}{1 + \alpha \Delta t} a_t$$

Substituting this into Eq. 26 gives the update for $z_t$:

$$z_t = z_{t-1} + \Delta t \left( \frac{1}{1 + \alpha \Delta t} v_{t-1} - \frac{\omega^2 \Delta t}{1 + \alpha \Delta t} z_{t-1} + \frac{\Delta t}{1 + \alpha \Delta t} a_t \right)$$

$$z_t = \left( 1 - \frac{\omega^2 \Delta t^2}{1 + \alpha \Delta t} \right) z_{t-1} + \frac{\Delta t}{1 + \alpha \Delta t} v_{t-1} + \frac{\Delta t^2}{1 + \alpha \Delta t} a_t$$

We can now write these two update rules in the standard LTI State-Space Model form $\mathbf{x}_t = \mathbf{A}\mathbf{x}_{t-1} + \mathbf{B}a_t$, where $\mathbf{x}_t = [z_t, v_t]^T$:

$$\mathbf{x}_t = \underbrace{\begin{bmatrix} 1 - \dfrac{\omega^2 \Delta t^2}{1 + \alpha \Delta t} & \dfrac{\Delta t}{1 + \alpha \Delta t} \\ -\dfrac{\omega^2 \Delta t}{1 + \alpha \Delta t} & \dfrac{1}{1 + \alpha \Delta t} \end{bmatrix}}_{\mathbf{A}} \mathbf{x}_{t-1} + \underbrace{\begin{bmatrix} \dfrac{\Delta t^2}{1 + \alpha \Delta t} \\ \dfrac{\Delta t}{1 + \alpha \Delta t} \end{bmatrix}}_{\mathbf{B}} a_t \tag{27}$$

The output equation is simply $z_t = \mathbf{C}\mathbf{x}_t$, with $\mathbf{C} = [1 \quad 0]$.

## C.3 Proofs of Propositions

**Proposition 5** (Equivalence to Causal Convolution). *The solution $z_t$ of the LTI system $\mathbf{x}_t = \mathbf{A}\mathbf{x}_{t-1} + \mathbf{B}a_t, z_t = \mathbf{C}\mathbf{x}_t$ can be expressed as a causal convolution of all past inputs.*

*Proof.* By unrolling the state-space recurrence relation, we get:

$$
\begin{aligned}
\mathbf{x}_t &= \mathbf{A}\mathbf{x}_{t-1} + \mathbf{B}a_t \\
&= \mathbf{A}(\mathbf{A}\mathbf{x}_{t-2} + \mathbf{B}a_{t-1}) + \mathbf{B}a_t \\
&= \mathbf{A}^2\mathbf{x}_{t-2} + \mathbf{A}\mathbf{B}a_{t-1} + \mathbf{B}a_t \\
&= \ldots \\
&= \mathbf{A}^t\mathbf{x}_0 + \sum_{m=0}^{t-1} \mathbf{A}^m\mathbf{B}a_{t-m}
\end{aligned}
$$

Assuming zero initial conditions ($\mathbf{x}_0 = \mathbf{0}$), the state is solely determined by the history of inputs:

$$
\mathbf{x}_t = \sum_{m=0}^{t-1} \mathbf{A}^m\mathbf{B}a_{t-m}
$$

Applying the output equation $z_t = \mathbf{C}\mathbf{x}_t$:

$$
z_t = \mathbf{C}\sum_{m=0}^{t-1} \mathbf{A}^m\mathbf{B}a_{t-m} = \sum_{m=0}^{t-1} (\mathbf{C}\mathbf{A}^m\mathbf{B})a_{t-m}
$$

We can extend the sum to infinity by defining the kernel $g[m] = \mathbf{C}\mathbf{A}^m\mathbf{B}$ for $m \geq 0$ and assuming a causal system where $a_k = 0$ for $k < 0$. This gives the convolution form:

$$
z_t = \sum_{m=0}^{\infty} g[m]a_{t-m}
$$

For a damped system, the spectral radius $\rho(\mathbf{A}) < 1$, ensuring the IIR filter is stable. $\qquad\square$

**Proposition 6** (FIR Approximation). *The IIR convolution can be approximated with arbitrary precision $\varepsilon$ by a Finite Impulse Response (FIR) filter of sufficient length $R$.*

*Proof.* The error introduced by truncating the infinite sum (the IIR filter kernel $g[m]$) at length $R-1$ is the tail of the sum:

$$
e_t = \left| \sum_{m=0}^{\infty} g[m]a_{t-m} - \sum_{m=0}^{R-1} g[m]a_{t-m} \right| = \left| \sum_{m=R}^{\infty} g[m]a_{t-m} \right|
$$

Let the input be bounded, $\|a_t\|_\infty \leq M_{in}$, and the matrix norms be bounded such that $\|\mathbf{A}^m\| \leq K\rho^m$ for some constants $K > 0$ and $0 < \rho < 1$ (guaranteed for a stable system). We can bound the error:

$$
\|e_t\|_\infty \leq \sum_{m=R}^{\infty} \|\mathbf{C}\|\|\mathbf{A}^m\|\|\mathbf{B}\|\|a_{t-m}\|_\infty
$$

$$
\leq \sum_{m=R}^{\infty} \|\mathbf{C}\|(K\rho^m)\|\mathbf{B}\|M_{in}
$$

$$
= KM_{in}\|\mathbf{C}\|\|\mathbf{B}\| \sum_{m=R}^{\infty} \rho^m
$$

The last term is a geometric series, which sums to $\frac{\rho^R}{1-\rho}$. Therefore:

$$
\|e_t\|_\infty \leq KM_{in}\|\mathbf{C}\|\|\mathbf{B}\|\frac{\rho^R}{1-\rho}
$$

To ensure the error is less than a desired precision $\varepsilon$, we require:

$$
KM_{in}\|\mathbf{C}\|\|\mathbf{B}\|\frac{\rho^R}{1-\rho} \leq \varepsilon
$$

Solving for $R$ gives the required receptive field length (filter size):

$$
R \geq \frac{\log\left(\frac{KM_{in}\|\mathbf{C}\|\|\mathbf{B}\|}{\varepsilon(1-\rho)}\right)}{\log(1/\rho)}
$$

This shows that a finite kernel length $R$ is sufficient to approximate the true physical dynamics to any desired precision. □

# D   GENERALIZATION THEORY OF PHASE-NET

**Problem Setup.**   Consider the stable linear time–invariant (LTI) system derived from the physics model:

$$\mathbf{x}_t = \mathbf{A}\mathbf{x}_{t-1} + \mathbf{B}a_t, \qquad z_t = \mathbf{C}\mathbf{x}_t = \sum_{m=0}^{\infty} g[m]\,a_{t-m}, \quad g[m] = \mathbf{C}\mathbf{A}^m\mathbf{B}.$$

In the network implementation we use a finite–length causal convolution. Let the temporal window length be $R$, define the input vector

$$\phi_t = (a_t, a_{t-1}, \ldots, a_{t-R+1}) \in \mathbb{R}^R,$$

and the truncated FIR coefficient vector

$$w = (g[0], g[1], \ldots, g[R-1]).$$

The predictor can be written as

$$f(\phi_t) = \langle w, \phi_t \rangle.$$

**Physical Facts.**   **Fact 1 (Stability).** Causality and spectral normalization guarantee $\rho(\mathbf{A}) < 1$. Hence there exist constants $K > 0$ and $0 < \rho < 1$ such that

$$\|\mathbf{A}^m\| \le K\rho^m, \quad \forall m \ge 0.$$

**Fact 2 (Magnitude and Norm Bounds).** The input amplitude is bounded by $M_{\text{in}}$. Weight regularization ensures $\|\mathbf{B}\| \le B_0$ and $\|\mathbf{C}\| \le C_0$. Therefore the $\ell_1$ norm of the convolution kernel satisfies

$$\|w\|_1 = \sum_{m=0}^{R-1} |g[m]| \le \sum_{m=0}^{\infty} C_0 K B_0 \rho^m = \frac{U}{1-\rho}, \qquad U \triangleq C_0 K B_0.$$

**Fact 3 (FIR Truncation Error).** Because $|g[m]| \le U\rho^m$,

$$\sum_{m=R}^{\infty} |g[m]| \le \frac{U\rho^R}{1-\rho}.$$

Since $\|a_t\|_\infty \le M_{\text{in}}$, the difference between the infinite IIR output and the length–$R$ FIR output satisfies

$$|z_t - z_t^{(R)}| \le \frac{U}{1-\rho} M_{\text{in}} \rho^R \triangleq \Gamma\rho^R.$$

This term can be made arbitrarily small by increasing $R$.

**Rademacher Complexity.**   Consider samples $\{\phi_i\}_{i=1}^n$ with $\|\phi_i\|_\infty \le M_{\text{in}}$. The empirical Rademacher complexity is

$$\widehat{\mathfrak{R}}_n = \mathbb{E}_\sigma \left[ \sup_{\|w\|_1 \le L} \frac{1}{n} \sum_{i=1}^n \sigma_i \langle w, \phi_i \rangle \right],$$

where $\sigma_i$ are independent Rademacher variables and $L = U/(1-\rho)$.

**Step 1 (Dual Norm Representation).** By $\ell_1$–$\ell_\infty$ duality,

$$\widehat{\mathfrak{R}}_n = \frac{L}{n} \mathbb{E}_\sigma \left\| \sum_{i=1}^n \sigma_i \phi_i \right\|_\infty.$$

**Step 2 (Bounding the Maximal Coordinate).** For any coordinate $j \le R$, the random variable $\sum_{i=1}^n \sigma_i \phi_{i,j}$ has magnitude at most $nM_{\text{in}}$. Khintchine–Kahane inequality together with a union bound yields

$$\mathbb{E}_\sigma \max_{1 \le j \le R} \left| \sum_{i=1}^n \sigma_i \phi_{i,j} \right| \le M_{\text{in}} \sqrt{2n \log(2R)}.$$

**Step 3 (Complexity Bound).** Substituting the above into the dual form gives

$$\widehat{\mathfrak{R}}_n \le L M_{\text{in}} \sqrt{\frac{2 \log(2R)}{n}}.$$

Taking expectation shows that the true Rademacher complexity satisfies

$$\mathfrak{R}_n \le \frac{U}{1 - \rho} M_{\text{in}} \sqrt{\frac{2 \log(2R)}{n}}.$$

**Source-Domain Generalization.** Let the loss $\ell$ be $L_\ell$–Lipschitz and bounded in $[0, 1]$. By the standard Rademacher generalization inequality, with probability at least $1 - \delta$ over the random draw of the training set,

$$\mathcal{E}_{\text{src}}(f) \le \widehat{\mathcal{E}}_n(f) + 2 L_\ell \mathfrak{R}_n + 3 \sqrt{\frac{\log(2/\delta)}{2n}} + O(\rho^R).$$

Plugging in the bound on $\mathfrak{R}_n$ gives

$$\mathcal{E}_{\text{src}}(f) \le \widehat{\mathcal{E}}_n(f) + O\left(\sqrt{\frac{\log R}{n}}\right) + O(\rho^R).$$

**Target-Domain Risk.** Let $\mathbb{P}_{\text{src}}$ and $\mathbb{P}_{\text{tgt}}$ denote the source and target distributions, and $W_1$ their 1–Wasserstein distance. Since $f$ is $L_f$–Lipschitz with

$$L_f \le \|w\|_1 \le \frac{U}{1 - \rho},$$

the discrepancy between source and target satisfies

$$\text{Disc} \le L_\ell L_f W_1(\mathbb{P}_{\text{src}}, \mathbb{P}_{\text{tgt}}) \le L_\ell \frac{U}{1 - \rho} W_1(\mathbb{P}_{\text{src}}, \mathbb{P}_{\text{tgt}}).$$

By the triangle inequality,

$$\mathcal{E}_{\text{tgt}}(f) \le \mathcal{E}_{\text{src}}(f) + \text{Disc}.$$

Combining with the source bound yields

$$\mathcal{E}_{\text{tgt}}(f) \le \widehat{\mathcal{E}}_n(f) + O\left(\sqrt{\frac{\log R}{n}}\right) + O(\rho^R) + L_\ell \frac{U}{1 - \rho} W_1(\mathbb{P}_{\text{src}}, \mathbb{P}_{\text{tgt}}).$$

**Choice of $R$.** To make the truncation error $O(\rho^R)$ smaller than the statistical term, choose

$$R \gtrsim \frac{2 \log n}{\log(1/\rho)} = \Theta(\log n).$$

With this choice, $\rho^R$ is negligible and the bound simplifies to

$$\mathcal{E}_{\text{tgt}}(f) \le \widehat{\mathcal{E}}_n(f) + O\left(\sqrt{\frac{\log \log n}{n}}\right) + L_\ell \frac{U}{1 - \rho} W_1(\mathbb{P}_{\text{src}}, \mathbb{P}_{\text{tgt}}).$$

**Comparison with Unconstrained Models.** For an unconstrained temporal model with hypothesis class $\mathcal{F}_{\text{base}}$, one typically has

$$\mathfrak{R}_n(\mathcal{F}_{\text{base}}) = O\left(\sqrt{\frac{C}{n}}\right),$$

where the capacity constant $C$ depends on depth, width, or spectral norm and is usually much larger than $\log \log n$. Thus the physics–informed class enjoys a strictly smaller statistical term $O(\sqrt{\log \log n / n})$ under the same sample size $n$.

## E   DETAILED DESCRIPTION OF ZAS

The Zero-FLOPs Axial Swapper (ZAS) is a lightweight spatial mixing operator designed to enrich long-range dependencies without adding computational burden. By selectively permuting a small subset of feature channels through block-wise transposition, ZAS introduces cross-region interactions that enhance the receptive field while keeping the temporal dimension untouched. Because the operation is purely an index reordering, it adds no learnable parameters and incurs zero FLOPs, making it ideal for efficiency-critical scenarios and stable gradient propagation.

---

**Algorithm 1** Zero-FLOPs Axial Swapper (ZAS)

---

Feature tensor $X \in \mathbb{R}^{B \times C \times T \times H \times W}$
Output tensor $\tilde{X} \in \mathbb{R}^{B \times C \times T \times H \times W}$
**Step 1. Channel partition.**
Split $X$ into two disjoint parts:

$$X = \begin{bmatrix} X_{\text{id}}, \ X_{\text{swap}} \end{bmatrix},$$

where $X_{\text{id}}$ contains the first $C - k$ channels and $X_{\text{swap}}$ contains the last $k = \lfloor pC \rfloor$ channels to be permuted.
**Step 2. Block partition.**
Given a block size $b$, crop the core region $H_2 = \lfloor H/b \rfloor \cdot b$, $W_2 = \lfloor W/b \rfloor \cdot b$, and reshape each spatial slice of $X_{\text{swap}}$

$$\mathcal{P} : \mathbb{R}^{H_2 \times W_2} \to \mathbb{R}^{\frac{H_2}{b} \times \frac{W_2}{b} \times b \times b}$$

into a grid of non-overlapping $b \times b$ blocks.
**Step 3. Block-wise transpose.**
For each $b \times b$ block $Z$, apply the inner transpose

$$\mathcal{T}(Z)_{u,v} = Z_{v,u}.$$

This operation is performed independently for every block and for all batches, channels, and time frames.
**Step 4. Reconstruction.**
Recover the spatial layout by the inverse partition

$$\text{ZAS}(X_{\text{swap}}) = \mathcal{P}^{-1}\big(\mathcal{T}(\mathcal{P}(X_{\text{swap}}))\big).$$

Concatenate with the unchanged channels to obtain the output:

$$\tilde{X} = \begin{bmatrix} X_{\text{id}}, \ \text{ZAS}(X_{\text{swap}}) \end{bmatrix}.$$

**Remark.**
ZAS performs only index reordering and introduces *zero learnable parameters* and *zero FLOPs*; its Jacobian is a permutation matrix, ensuring gradient safety and perfect energy preservation.

---

## F    ROBUSTNESS TO LIGHTING VARIATIONS

To further evaluate cross-illumination robustness, we measure the mean absolute error (MAE, bpm) of PHASE-Net, PhysNet, and RhythmFormer under four representative lighting settings (Fig. 5). PHASE-Net consistently achieves the lowest error across all conditions—4.15/3.80/2.70/4.31 bpm for LED-Low/High/Incandescent/Nature—substantially outperforming RhythmFormer (5.85/4.46/3.64/5.65 bpm) and PhysNet (18.04/17.59/16.17/21.62 bpm). In particular, PHASE-Net maintains strong accuracy in the challenging *Incandescent* and *Nature* settings, demonstrating superior generalization to complex illumination and outdoor reflectance. These results confirm that PHASE-Net offers a tighter error bound and greater stability for real-world deployment under diverse lighting conditions.

## G    VISUALIZATION OF THE PREDICTED AND GROUND-TRUTH BVP

We randomly select representative clip samples from the UBFC-rPPG Bobbia et al. (2017) and PURE Stricker et al. (2014) datasets and visualize both the predicted rPPG waveforms and their corresponding power spectral density (PSD) curves in Fig. 6 and Fig. 7. These qualitative results provide an intuitive view of model behavior: the predicted signals not only closely follow the ground-truth BVP in amplitude and phase but also exhibit highly consistent dominant frequency peaks in the PSD domain, indicating accurate heart-rate estimation. Across both controlled (PURE) and more unconstrained (UBFC) scenarios, PHASE-Net preserves the fine-grained temporal structure of the pulse waveform and maintains sharp, well-aligned spectral peaks, further validating its ability to recover clean physiological rhythms despite variations in illumination, motion, and sensor noise.

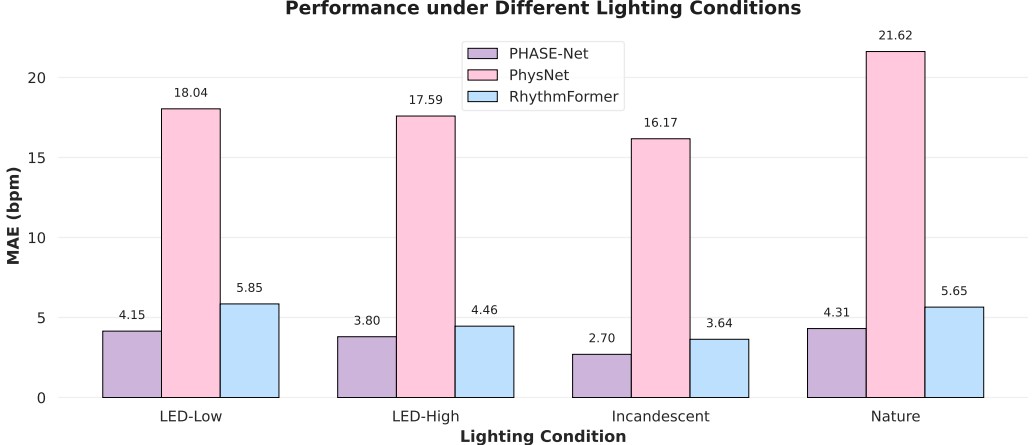

**Figure 5:** MAE (bpm) of PHASE-Net, PhysNet, and RhythmFormer under four lighting conditions: LED-Low, LED-High, Incandescent, and Nature. Lower is better.

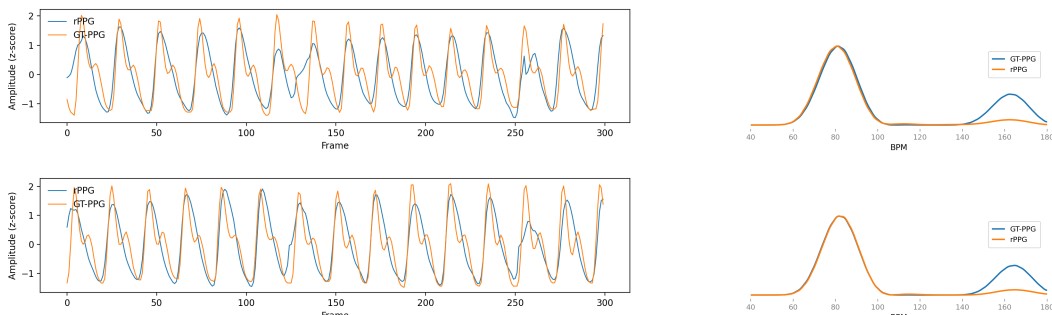

**Figure 6:** Visual comparison of the rPPG signals (left) predicted by PHASE-Net and their corresponding PSDs (right), alongside the respective ground-truth in PURE Stricker et al. (2014).

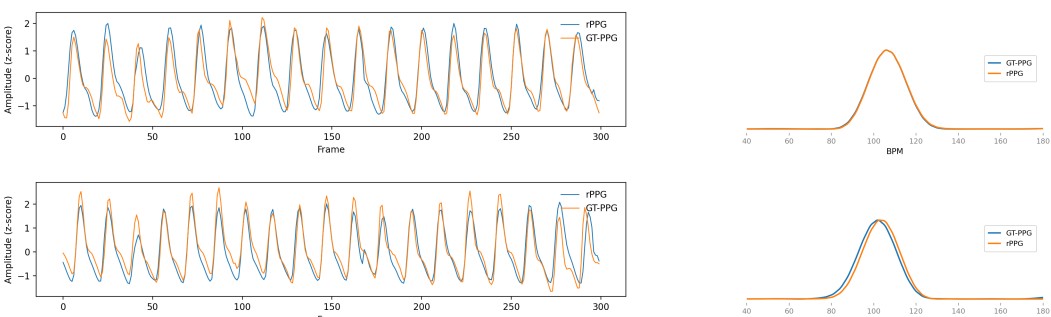

**Figure 7:** Visual comparison of the rPPG signals (left) predicted by PHASE-Net and their corresponding PSDs (right), alongside the respective ground-truth in UBFC-rPPG Bobbia et al. (2017).

