# OpenReview forum: "PHASE-Net: Physics-Grounded Harmonic Attention System for Efficient Remote Photoplethysmography Measurement"
_ICLR.cc/2026/Conference — ICLR 2026 Conference Withdrawn Submission_

### Official Review · Reviewer_xenY · 2025-10-23

**Soundness:** 2
**Presentation:** 2
**Contribution:** 2
**Rating:** 4
**Confidence:** 5

**Summary:**

Remote photoplethysmography (rPPG) enables non-contact physiological monitoring but suffers from accuracy loss under motion/illumination changes, with existing deep learning methods lacking theoretical grounding. This work proposes a physics-informed rPPG paradigm derived from hemodynamics’ Navier–Stokes equations, showing pulse signals follow a second-order dynamical system—justifying Temporal Convolutional Network (TCN) use. It designs PHASE-Net, a lightweight model with three key modules: Zero-FLOPs Axial Swapper, Adaptive Spatial Filter, and Gated TCN. Extensive experiments confirm PHASE-Net achieves state-of-the-art performance, offering a theoretically sound, deployment-ready rPPG solution.

**Strengths:**

1. A large number of methods were compared and experiments were conducted on a large number of datasets.
2. Clear and precise expression of the method.
3. The algorithm is lightweight and highly efficient.

**Weaknesses:**

1. Innovation is very limited, and the modified online methods are very common.
2. Many STMap-based methods lack comparisons, and those methods seem to be more lightweight.
Dual-gan: Joint bvp and noise modeling for remote physiological measurement
RhythmNet: End-to-end Heart Rate Estimation from Face via Spatial-temporal Representation
3. Many of the latest cross-domain papers do not include comparisons.
Neuron Structure Modeling for Generalizable Remote Physiological Measurement
Dual-bridging with adversarial noise generation for domain adaptive rppg estimation
4. There were no evaluations conducted in challenging scenarios such as VIPL-HR and V4V. The dataset used in the paper is already very saturated, and the evaluation value is relatively low.

**Questions:**

The problem is elaborated in the drawbacks, mainly due to the lack of innovation.

---

### Official Review · Reviewer_zZa9 · 2025-10-28

**Soundness:** 3
**Presentation:** 2
**Contribution:** 2
**Rating:** 4
**Confidence:** 3

**Summary:**

This paper presents PHASE-Net, a lightweight and theoretically grounded deep learning model for remote Photoplethysmography measurement. The core innovation lies in establishing a physical connection between the intrinsic dynamics of the rPPG signal and a specific neural network architecture. The authors rigorously derive the underlying physiological dynamics from the Navier-Stokes equations of hemodynamics, simplifying it to a second-order Damped Harmonic Oscillator Ordinary Differential Equation. Crucially, they prove that the discrete solution of this ODE is equivalent to a causal convolution, thus providing a solid theoretical basis for employing a TCN. PHASE-Net integrates three key modules: a Gated TCN for modeling the physics-driven pulse dynamics, a Zero-FLOPs Axial Swapper for zero-cost spatial feature mixing, and an Adaptive Spatial Filter for dynamically focusing on signal-rich regions and explicitly encoding local pulse velocity. Extensive experiments demonstrate that PHASE-Net achieves state-of-the-art performance on multiple public datasets, showing excellent cross-domain generalization and high computational efficiency.

**Strengths:**

1. On datasets such as UBFC-rPPG and PURE, PHASE-Net achieves SOTA performance in terms of MAE and RMSE metrics, with high waveform fidelity.

2. The model attains SOTA performance with extremely low computational and parameter costs (0.29M parameters, 28.3G MACs).

**Weaknesses:**

1. The paper claims to be a "physics-grounded" model; however, only the GTCN module is directly derived from physical principles (ODE → TCN). The ZAS and ASF modules are essentially engineering optimizations (addressing spatial heterogeneity and feature mixing). While they contribute to performance, they do not appear to have a direct, formal connection to the physics of the damped harmonic oscillator ODE.

2. Although the theoretical derivation is innovative, the performance comparison with recent methods (such as PhysMamba, RhythmMamba, etc.) needs to be more comprehensive. The paper should more clearly articulate in which specific aspects it surpasses these state-of-the-art methods.

3. ZAS is shown to be effective, but it is essentially a spatial permutation operation. The ablation studies lack a comparison with traditional, albeit computationally costly, modules also used for spatial mixing, such as 1x1 convolutions, simpler feature rearrangement, or channel shuffling as in ShuffleNet.

4. There are some typos and formatting issues that should be corrected:

(1) In Table 4, "MDNet(Ours)" is likely a typo; the proposed method should be named PHASE-Net.

(2) Line 182: The notation 𝑧(𝑡) :=𝑝′(𝑥0,𝑡)should be checked for consistency.

(3) Formatting inconsistencies exist, for example, lines 222 and 238.

**Questions:**

Please refer the above weaknesses.

**Details Of Ethics Concerns:**

None.

---

### Official Review · Reviewer_JCsD · 2025-10-30

**Soundness:** 2
**Presentation:** 2
**Contribution:** 2
**Rating:** 4
**Confidence:** 4

**Summary:**

The paper proposes PHASE-Net, a physics-grounded rPPG model for non-contact physiological measurement. Starting from a linearized hemodynamic formulation, the authors reduce Navier–Stokes to a forced damped harmonic oscillator at a fixed facial point and show its discrete LTI solution is a causal convolution, motivating a Temporal Convolutional Network (TCN) backbone. Around this, they add (i) Zero-FLOPs Axial Swapper (ZAS) to mix spatial regions via reversible permutations, (ii) an Adaptive Spatial Filter (ASF) that produces framewise spatial masks and concatenates a first-order temporal derivative, and (iii) a gated, dilated TCN (GTCN).

**Strengths:**

•  The paper proposes some theoretical ways into rPPG measurement problems.
•  Robust generalization. Strong leave-one-dataset-out results, especially for PURE and BUAA.

**Weaknesses:**

- “Novelty” of LTI→convolution→FIR. The propositions are correct but classical; the novelty lies in the application and design. Consider acknowledging prior control/signal-processing results explicitly and sharpening what is new for rPPG. The reason and motivation about why this design can improve efficiency is unclear.

- Disconnection between theory and some modules: Some modules, including zero-flops axial swapper and adaptive spatial filte,r are not related to the theory in Sec. 3.1. The motivation to design these two modules is not clear.

- Broader baselines. You compare to recent CNN/Transformer/Mamba-style methods, but accuracy tables could include more state-space baselines beyond PhysMamba/RhythmMamba, where possible, and a plain TCN without physics knowledge to isolate the value of the derivation.

- Impulse response & interpretability. Since the model approximates an FIR, can you visualize learned impulse responses and peak frequencies per subject/clip to verify physiological plausibility?

**Questions:**

Please see the weakness part.

---

### Note · Authors · 2025-11-13

I have read and agree with the venue's withdrawal policy on behalf of myself and my co-authors.